# Spectrum, Time Course, Stages, and a Proposal for the Diagnosis of Histamine Intolerance in General Practice: A Nonrandomized, Quasi-Experimental Study

**DOI:** 10.3390/jcm14020311

**Published:** 2025-01-07

**Authors:** József Tamasi, László Kalabay

**Affiliations:** 1Department of Family Medicine, Semmelweis University, 1085 Budapest, Hungary; 2Department of Internal Medicine and Hematology, Semmelweis University, 1088 Budapest, Hungary

**Keywords:** histamine, histamine intolerance, mast cell activation syndrome (MCAS)

## Abstract

**Background/Objectives:** Limited research has explored histamine intolerance from the perspective of primary caregivers. Our objective was to develop a practical symptom profile from the standpoint of general practice. We also aimed to gather data on the frequency and timing of disease progression and to establish a staging system. **Methods:** This study utilized a nonrandomized, quasi-experimental design. An in-depth interview was conducted with 217 patients involving 120 questions. To evaluate associations between food intake and symptoms, we recommended either an exclusion diet or a low-histamine diet. A follow-up questionnaire was subsequently administered. We also analyzed 3831 doctor–patient meetings involving upper respiratory symptoms. **Results:** Symptoms in 77 patients were associated with histamine-rich meals. The most characteristic symptoms included respiratory symptoms (95%), bloating (94%), headache (91%), fatigue (83%), postprandial drowsiness (81%), skin symptoms (81%), diarrhea/loose stool (77%), psychological symptoms (77%), dyspepsia (69%), and muscle/eyelid twitching (61%). Patients with suspected histamine intolerance visited primary care three times more often with upper respiratory symptoms than those without suspected histamine intolerance. The symptom spectrum of histamine intolerance involves multiple organ systems and occurs in distinct, repeating patterns. Symptoms can be described by their duration, sequence, and severity level, which is the key focus of this research, including visual representations. In its most severe stages, histamine intolerance may potentially involve mast cell activation. A personalized diet is associated with a gradual reduction in both the intensity and frequency of symptoms. **Conclusions**: The spectrum of histamine intolerance can be characterized by specific symptom patterns with defined frequencies, timelines, and symptom stages.

## 1. Introduction

Histamine intolerance (HIT) is best described as a multisystemic disorder displaying an intermittent, altering, and individual symptom spectrum. The multifaceted clinical picture is due to the extensive utilization of histamine on four receptors. The multifunctionality of this specific biogenic amine poses a challenge in the everyday diagnostic workup of patients exhibiting symptoms of several organ systems in an inconsistent manner [1].

Diamine oxidase (DAO) and hydroxy-N-methyl transferase (HNMT) are responsible for the breakdown of exogenous histamine. Other biogenic amines also interfere with its decomposition, but in everyday meals, histamine is present in amounts high enough to cause widespread effects in sensitive patients [2]. For these reactions to occur, there must be a deficiency of breakdown (due to genetic causes, etc.).

The “German guideline for the management of adverse reactions to ingested histamine” recommends referring to HIT as an “adverse reaction to ingested histamine” [3]. It also suggests that applying a low-histamine diet currently represents the only, yet insufficient, diagnostic method. Diagnostic tests for histamine intolerance, including serum DAO concentration, DAO activity, and urinary histamine metabolites, face significant challenges due to the heterogeneity of the condition’s pathophysiology, difficulties in implementation, and lack of specificity. This greatly complicates the objective monitoring of HIT.

The involvement of histamine and mast cells has been a subject of research in numerous primary diseases, including primary headaches, irritable bowel syndrome (IBS), bronchial asthma, and dyspepsia [4,5]. Furthermore, some of these show co-occurrence [6,7,8,9,10,11,12], and HIT may be an underlying factor when multiple primary diseases co-occur.

Foods high in histamine can provoke symptoms not only in HIT but also in mast cell activation disorder (MCAD) and mast cell activation syndrome (MCAS). MCAD refers to a broader spectrum of MCA related diseases, while MCAS is a diagnosis of exclusion, similar to HIT. However, MCAS cannot be confused with HIT; it is supported by validated laboratory markers [13,14]. Among the markers for MCAS, tryptase level is considered the most reliable, with a diagnostic criterion of baseline +20% + 2 ng/mL 1–4 h after symptom onset. Other markers, such as urinary prostaglandin D2 metabolites, histamine metabolites, and plasma histamine, can only serve as supportive evidence due to the lack of validated thresholds [15]. On the other hand, MCAS and HIT share a laboratory marker: urinary N-methylhistamine.

Overlapping aspects of HIT and MCAD are particularly evident in clinical symptoms, which can include allergy-like upper respiratory issues, IBS-like symptoms, dermatological symptoms (e.g., flushing), asthma-like symptoms, cardiovascular symptoms, psychological symptoms, gastric hypersecretion, vomiting, and headaches, among others. At present, it is unknown whether this overlap is attributed to shared genetic factors or other hypothetical causes, such as histamine-induced activation of mast cells.

In our research, we adopted the broader perspective of “Consensus-2” for MCAD definitions [16], as opposed to the narrower definitions provided by “Consensus-1” [17,18]. Certain sources even estimate that MCAD might affect as many as 17% of the population [19]. The “Consensus-2” guideline recommends considering comorbidities as potential links to MCAS. A possible genetic link could be mutations in DAO or HNMT.

Our study focused on establishing clinical symptom patterns in HIT’s presentation spectrum from the perspective of a primary caregiver. Furthermore, our goal was to observe the chronological development of symptoms in order to understand their timeline and frequency of onset. We sought to establish patterns in symptom development and regression by describing stages of symptom severity. We attempted to develop a questionnaire with items that align with the pathophysiology of HIT, as the diagnosis of HIT in general practice is challenging due to the lack of a validated laboratory marker. Finally, our objective was to determine the extent to which the condition could be influenced by diet across different levels of severity and to identify indicator symptoms.

## 2. Methods

### 2.1. Study Design

This study, designed as a nonrandomized, quasi-experimental study, involved three doctor–patient meetings. During the first visit, we utilized a questionnaire to explore relevant symptoms of suspected HIT. In the second meeting, patients returned with a food–symptom diary and reported whether they found a connection between their diet and their symptoms. Therefore, we were able to establish a confirmed HIT diagnosis in 77 patients. If so, a low-histamine diet was implemented. During the third visit, we reviewed the observations related to the low-histamine diet.

Conducted between 26 September 2023 and 30 September 2024 in a large town setting, recruitment relied on online platforms, posters, and verbal invitations encouraging patients with primary complaints to explore the option of scheduling appointments. Appointments for all three doctor–patient meetings could be scheduled via a designated phone number and email address. The primary objective was to evaluate associations between a low-histamine diet and symptom changes, while the secondary objective was to gather information about the disease to support and improve diagnostic measures.

A total of 244 patients with chronic symptoms were invited, and 217 agreed to be interviewed. Patients with primary symptoms of headache and/or diarrhea were enrolled. Exclusion criteria involved a secondary origin of symptoms from any other condition (e.g., gluten sensitivity, food allergy, inflammatory bowel disease, etc.), with the exception of self-reported lactose intolerance. No medication was given during the examination. The questionnaire contained 120 questions. All questions and answers are included in Appendix A. To capture some of the essential findings from our study, four original figures were generated. Data collection took place prospectively, covering a follow-up period of 3–6 months.

Suspected HIT patients were asked to maintain a food and symptom diary. The objective was to carefully observe and evaluate associations between their meals and their experienced symptoms. In instances where they identified a potential link between their meals and their symptoms, a second visit was arranged to discuss further steps. During this meeting, patients were advised to follow either a low-histamine or an elimination diet, each for a period of 3–6 weeks. If any improvement was noted as a result of these diets, the diagnosis of confirmed HIT was established, and a third meeting was arranged to carry out a final assessment utilizing a 20-question questionnaire.

The elimination diet focused on the exclusion of high-histamine foods on the basis of the patients’ food–symptom diary. On the other hand, a low-histamine diet was based on fresh fruits, vegetables, potatoes, rice, and fresh poultry. These diets were not gluten-free and did not exclude dairy products. Moreover, the excluded foods were later allowed to be consumed in small amounts according to personal tolerance levels.

Patients were also provided very detailed educational material about the disease and its dietary management. A portion of the supplied material contained questions to facilitate the maintenance of the food and symptom diary and help link symptoms to meals (Appendix A). These identical questions were subsequently utilized to fine-tune the dietary regimen. The approach of fine-tuning with certain indicator symptoms is novel, as is the utilization of many of these questions.

The questionnaire focused on the presence of possible concurrent symptoms that would raise suspicion of HIT. We asked whether the patient had experienced any of them with a certain degree of regularity (at least 1–2 times/6 months) for some of the past years. If the patient had already been on an exclusion diet, the presence of symptoms prior to the restrictions was taken into account. Questions were occasionally repeated in a rephrased form to ensure the accuracy of the response.

The statistical analysis was performed with IBM SPSS Statistics 25 (Armonk, NY, USA). A value of *p* < 0.05 was considered as significant.

This study was approved by the Ethics Committee of the Hungarian National Institute of Public Health and the Department of Clinical Research at the National Institute of Pharmacy (OGYÉI/54189-4/2023) on 26 September 2023.

### 2.2. Establishment of a HIT Symptom Frequency Score

Symptoms occurring less than 1 day/month were assigned 1 point, symptoms occurring 1–6 days/month (1 day/week) were given 2 points, symptoms occurring 7–14 days/month (2–4 days/week) received 3 points, symptoms occurring 15–21 days/month (4–5 days/week) were assigned 4 points, and symptoms occurring mostly every day were given 5 points. A subjective symptom severity score was also established, where 10 represented the most severe symptoms.

### 2.3. Analysis of Doctor–Patient Meetings Involving Upper Respiratory Symptoms

We also collected data on patients visiting primary care with the following ICD-10 diagnoses: B34.9 (viral infection), J06.9 (acute upper respiratory infection), J03.9 (acute tonsillitis), and J20.9 (acute bronchitis). Our hypothesis was that suspected HIT patients visit primary care more frequently than other patients. A total of 7083 doctor–patient meetings were examined from 1 September 2021 to 27 September 2024. Meetings occurring within a one-week timeframe were counted as one, and patients with chronic obstructive bronchitis were excluded. This left 3831 primary care meetings. We only included patients who had at least one meeting with one of the four previously mentioned diagnoses; those with none were not counted.

## 3. Results

### 3.1. Formation and Symptom Frequency of HIT and Non-HIT Patient Groups

Two subgroups were created from the 217 suspected HIT patients. The first subgroup contained 77 confirmed HIT patients (35.5% of all patients), who experienced some improvement in response to a low-histamine or exclusion diet. The second subgroup comprised 140 patients (64.5%) without HIT. The baseline characteristics of these groups are shown in Table 1.

The most relevant symptoms within the subgroup of confirmed HIT patients are included in Table 2. Ninety-nine percent of HIT patients had already identified or avoided at least one food or beverage triggering one or more of their symptoms before receiving the formal diagnosis. The most frequently listed items are included in Table 2.

For a statistical comparison of symptoms between the HIT and non-HIT subgroups, see Table 3. In the non-HIT subgroup, only 16 patients kept a food and symptom diary, while the rest self-reported that there was no association between food intake and their symptoms.

Sixty-six percent of surveyed HIT patients reported feeling more energetic when they had an empty stomach. A significant majority (96%) confirmed that they did not experience most or any symptoms during the second or third trimester of pregnancy, except for occasional cases of dyspepsia. Eighty-seven percent of HIT individuals could already associate their symptoms occurring intermittently with meals in various ways at the time of the first interview. Sixty-two percent experienced the highest intensity of symptoms in the late afternoon or evening, particularly following the two main meals. Furthermore, symptoms frequently subsided during the overnight fasting period except in instances when dinner triggered morning symptoms.

Half of the HIT respondents had children aged 6 to 39 years who displayed somewhat similar symptoms. A total of 70% had a family member suffering from symptoms of diarrhea and/or headache, 51% had a relative who took medication for dyspepsia, and 17% had a family member with HIT. Twelve percent of individuals even concurrently experienced headaches with their close relative.

A subset of patients voluntarily underwent testing for DAO levels, as measuring DAO levels free of charge is not available in our country. Furthermore, the determination of DAO activity is entirely unavailable. A total of 74% (37/50) of all patients had DAO levels below the limit (10 kU/L), with HIT patients showing an even greater percentage (82%, 28/34) of positive results (Table 4).

### 3.2. Analysis of Predictors for HIT

In the confirmed HIT subgroup, only 22% had a body mass index (BMI) exceeding the national average. A logistic regression model was used to distinguish between the presence and absence of HIT on the basis of four predictors: patient BMI, HIT symptom frequency score, onset of symptoms can be attributed to meals, and average number of nontolerated foods (χ^2^(4) = 43.2, *p* < 0.001). Binary logistic regression was then performed to generate an ROC curve, resulting in an AUC of 0.76 (*p* < 0.001, *n* = 217). When the DAO concentration was included as an additional predictor, the AUC increased to 0.794 (*p* = 0.001, *n* = 50).

### 3.3. Analysis of the Second Questionnaire Data

Following dietary adjustments, we utilized a second questionnaire to assess improvements in the frequency and subjective severity of symptoms, which were found to be significant (Table 4). Examples of symptom reduction include loose stools instead of diarrhea and moderate, dull headaches or only the aura phase in place of migraine-like symptoms. We excluded five HIT patients from the before–after score comparison because of their nonadherence to a diet for at least 4 weeks. A total of 72 patients (94%) still followed a diet, 47/77 (61%) an elimination diet, and 25/77 (32%) a low-histamine diet. We aimed to determine the time frame within which symptoms in the case of a dietary mistake typically emerged (on average 1.1 h), assess whether sudden lethargy was concomitant with HIT symptoms (40/77, 52%), examine whether symptom severity was observed to vary with the quantity of food consumed (52/77, 68%), and investigate whether the presence of an intercurrent disease exacerbated these symptoms (35/40, 88%). Overall mood improvements were observed in 29/77 (38%) participants following the diet.

### 3.4. General Trends Based on the Food and Symptom Diaries

To illustrate general trends from the food and symptom diaries, Figure 1, Figure 2, Figure 3 and Figure 4 were created. Figure 1 shows that the symptoms can be grouped by severity and elapsed time. The stages based on severity are shown in Figure 2. The grouping by time is presented in Figure 3 and Figure 4. Figure 3 illustrates the order in which immediate symptoms appear after consuming a specific meal. Figure 4 depicts the timeline of symptoms over the course of several days.

Figure 2 illustrates the gradual progression of the symptom spectrum of HIT. Each stage includes all the previous ones. Stage 1 contains the most common initial symptoms for every patient. Stage 2 was observed in 15/36 (42%), Stage 3 in 17/36 (47%), and Stage 4 in 4/36 (11%) patients where data were available.

Figure 3 presents a typical timeline of the immediate symptom occurrence of HIT following a single meal. This is likely due to the progression and absorption of food through the digestive system. The first symptoms are upper respiratory symptoms, followed by gastrointestinal symptoms, and then general and neurologic symptoms occur. For example, runny nose and sneezing initially appear, followed by early satiety and abdominal discomfort, and finally, postprandial somnolence and a dull headache.

Figure 4 contains four timelines that illustrate typical examples of HIT symptom recurrence over several days. Each patient experiences two to four of these timelines. Two types of timelines were observed in 7/36 (20%), three types of timelines in 13/36 (36%), and four types of timelines in 16/36 (44%) confirmed HIT patients, where data were available.

Symptoms of HIT can be described based on their duration, with typical timelines depicted in Figure 4: short and mild symptoms (Timeline 1), moderate duration (Timeline 2, lasting up to 3 days), and severe symptoms (Timeline 4). Some individuals experience infrequent, mild symptoms with short durations that are likely to occur in the late afternoon or evening after meals (Timeline 1). Moderate symptoms are followed by a relatively shorter period of extended symptoms (Timeline 2). The most severe cases may hypothetically be linked to MCA, resulting in a severe outbreak (Timeline 4) that subsides over a period of up to 2 weeks. More frequent lone surges of lower-level symptoms can sometimes indicate the imminent onset of such an outburst.

Continuous symptom persistence may also occur (Figure 4, Timeline 3). This may present as a symptom conglomerate or as an isolated symptom, such as coughing or acid reflux, that endures for an extended period and recurs or intensifies after meals. Likewise, some cases of headache appear to accumulate, worsening with each meal and persisting for several days. Even as the pain subsides, it may reappear after eating certain foods.

HIT symptoms can also be classified by severity, ranging from mild to severe (Figure 2, Stages 1 to 4). Patients with more severe symptoms tend to experience recurrences more frequently. More severe symptoms tend to manifest as symptom conglomerates, unlike those of Stage 1, which may appear alone. The initial stages are reversible (Stages 1–3a), whereas the more severe symptoms are not (Stages 3B and 4); they are only ameliorated through fasting or a low-histamine diet. Severe symptoms coincide with loss of appetite and possibly lead to skipping subsequent meals. Fasting cannot stop the prolonged stage of a severe outburst (Figure 4, Timeline 4) lasting 1–2 weeks; it can only mitigate recurring exacerbations, such as recurrent migraines or nausea.

Data for HIT symptom stages (Figure 2) and HIT timelines (Figure 4) are incomplete, as regular patterns were identified during the course of this study. Where feasible, we supplemented the questionnaire with additional data collected either retrospectively or prospectively (*n* = 36 instead of *n* = 77).

### 3.5. Upper Respiratory Symptoms Among HIT Patients

Since we focused on clinical features rather than obtaining laboratory results for MCAS, we could only classify our cases as suspected MCAD. A theoretical possibility for MCA was considered in instances where either recurring sore throat (with prolonged symptom persistence of common cold-like symptoms) or flushing occurred, totaling 41 of 77 confirmed HIT cases. The frequency and severity of these most intense symptoms varied widely. There was a significant difference in the average count of gastrointestinal (*p* = 0.005, Mann–Whitney test), neurological (*p* = 0.002), and respiratory (*p* = 0.013) symptoms, symptom severity score (*p* = 0.007), and average count of all complaints (*p* = 0.006) between the subgroup of patients with possible MCA (41 patients) and the remaining confirmed HIT patients (36 patients). Similar differences were found between these two groups for nausea (*p* = 0.016, Fisher’s exact test) and vomiting (*p* = 0.009).

We also examined 141 suspected HIT patients and 945 non-HIT patients for their primary care visits (3081) with upper respiratory symptoms. Suspected HIT patients had a median of six (Q1–Q3 = 3–8) doctor–patient meetings, while non-HIT patients had a median of two (Q1–Q3 = 1–4), with a significant difference (*p* < 0.001, Mann–Whitney).

## 4. Discussion

Few studies have documented the complaints observed at the initial presentation in patients with suspected HIT, especially in primary care settings. Such a survey has not yet been conducted in Hungary (provoking factors are likely different from those in other regions of the world). The occurrence rates and the large number of combinations of symptoms are comparable to those reported in other similar studies [20].

There is a connection between primary symptoms and dietary triggers [21,22,23,24]. Currently, there are no validated dietary guidelines for HIT [25,26]. The same holds true for MCAD, despite the expectation that these conditions would ideally share common elements. Furthermore, the exclusion of non-low-histamine foods in some HIT diets is unclear but may be due to overlap with MCAS. Additionally, the signs of HIT vary depending on the specific food matrix consumed [27,28].

### 4.1. Symptom Spectrum of HIT

Questions to help both patients and physicians understand connections were introduced in our questionnaire, as the diagnosis is challenging [29]. Traits such as feeling more energetic on an empty stomach and the onset of symptoms in the afternoon and evening, which are typically associated with main meals, suggest that food triggered the symptoms. The presence of allergy-like respiratory symptoms without known allergies further raised suspicion of HIT. A positive family history of similar symptoms was frequently encountered in our patients, indicating a genetic basis [30,31].

Some HIT symptoms manifest not only acutely but also over extended periods. These symptoms include ongoing upper respiratory issues, asthma-like complaints, dyspepsia, skin problems, dizziness, and psychological symptoms, among others. These will complicate differential diagnosis because of their delayed onset after meals and slow improvement. However, our results indicate that the majority of patients with suspected HIT seek medical attention for headaches.

Onset of HIT symptoms at a young age and abdominal pain during childhood were frequent [32]. Alleviation of most symptoms during the second and third trimesters in pregnancy (24/25 cases) may be attributed to the placenta’s significant increase in serum DAO by several hundredfold [33,34]. This increase compensates for the genetically impaired function of DAO, HNMT, or both.

We did not find a significant difference in most categories between confirmed HIT cases and the remaining patients. Reasons for nonsignificance may include the need for patients to have a higher level of basic knowledge about the disease, the requirement for high compliance, different pathophysiological backgrounds beyond mutations in DAO and HNMT, and the presence of other triggers, such as those seen in MCAD. In many cases, potential links with food intake were still present but not to the extent that HIT could be diagnosed.

Symptoms (Figure 2, Stage 1), such as postprandial somnolence/fatigue/brain fog, postnasal dripping, abdominal discomfort, loose stools, and muscle twitching, could serve as early indicators of dietary errors and could be used to identify suspected foods. On the other hand, the most severe symptoms (Figure 2, Stages 3–4), such as vomiting, migraines, and flushing, were the first to occur less frequently with dietary intervention. Postprandial drowsiness has not received enough attention, despite being a prevalent and characteristic symptom of HIT. Lapses in a low-histamine diet often trigger this symptom, making it a useful indicator for dietary adjustments.

A challenge in tracking symptoms was the variability in the time between the start of eating and the onset of symptoms. In general, more intense complaints tended to manifest with a shorter delay, and milder complaints often lasted for shorter intervals. The symptoms and timeline of HIT can also be compared with those observed during oral histamine provocation [35].

Indications that HIT is an indivisible disease are supported by the observation that as patients began adhering to their diets, less severe symptoms emerged across all affected areas. To estimate the decrease in disease activity, we utilized subjective symptom severity scores, frequency scores, and improvement in symptom severity stages. The most severely affected patients demonstrated the most noticeable regression in severity stages. However, if triggering foods were repeatedly ingested at a lower stage, HIT worsened, progressing to a higher stage.

Recent studies have highlighted the role of gut microbiota dysbiosis in HIT, with an overabundance of histamine-secreting bacteria such as *Staphylococcus*, *Proteus*, *Enterococcus faecalis*, and *Clostridium perfringens*, alongside a reduced presence of gut-health-associated bacteria like *Ruminococcus*, *Prevotellaceae*, *Faecalibacterium*, and *Faecalibacterium prausnitzii* [36]. Dietary interventions, including low-histamine diets and DAO supplementation, have been shown to reduce histaminogenic bacteria (e.g., *Pseudomonadaceae*, *Morganellaceae*, *Proteus mirabilis*, and *Raoultella*) and increase beneficial species such as *Roseburia* spp. [37]. These changes in gut microbiota composition may contribute to the observed improvement in symptomatology.

### 4.2. Severe Symptoms Associated with HIT

Some patients developed a more severe symptom conglomerate. MCA may be a possible explanation for these episodes [38,39]. This presentation may, in part, have resembled an upper respiratory infection with a cluster of symptoms that manifested in different forms (Figure 4, Timeline 4). This cluster of symptoms was characterized by sore throat, vomiting, migraine-like headaches, sometimes flushing, and other symptoms, as shown in Figure 2 (Stages 3B and 4). A cyclical recurrence occasionally occurred as often as every 3–6 weeks, while the laboratory inflammatory parameters (C-reactive protein, erythrocyte sedimentation rate, white blood cell count) remained low when tested. Each episode typically followed a consistent sequence in this order: malaise, sore throat, a single day (or a few hours) of elevated body temperature (consistent with its resemblance to an outburst rather than several days in case of an infection), flushing, and then 7–14 days of upper respiratory symptoms, potentially accompanied by coughing [40]. Interestingly, this symptom set varied in severity, and dietary modifications helped reduce symptom severity incrementally.

Given the genetic differences between HIT and certain types of MCAD, it is hypothetically possible that less severe cases of HIT could be associated with mutations of DAO and HNMT, which may respond well to an elimination diet. On the other hand, more severe cases could involve idiopathic MCAS, necessitating a more disciplined low-histamine diet. This hypothesis requires further investigation.

A few of the confirmed HIT patients had elevated urine histamine and N-methylhistamine results. This meant that according to the global “Consensus-2” criteria, a diagnosis of idiopathic MCAS could have been supported in these patients. Mast cell mediators, similar to histamine in HIT, are typically short-lived, difficult to measure, and often not available through commercial testing. “Consensus-2” advocates that while laboratory tests are essential in the diagnosis, efforts to improve patient care in regions where such tests are unavailable should not be hindered by the current limitations in testing given the relatively recent recognition of the disease.

Our questionnaire had some overlapping aspects with the Mast Cell Mediator Release Syndrome Questionnaire [41]. Idiopathic MCA is further suggested by the fact that our patients with concurrent illnesses experienced more severe HIT symptoms (35/40 HIT patients).

Our results indicated that patients with suspected HIT visited primary care three times more frequently for upper respiratory symptoms than non-HIT patients. This may have resulted from MCA, among other potential causes, including the possibility that their symptoms were more severe. If patients who did not visit the doctor at all had been included, the observed differences would have been even greater.

### 4.3. Proposed Diagnostic Steps for HIT

Creating a standardized screening questionnaire for HIT would be beneficial. The questions could include inquiries about intolerance to specific medications known to act as DAO blockers, such as clavulanic acid and acetylcysteine [42]. Twenty-one percent of the confirmed HIT patients were sensitive to one of these medications. Moreover, self-reporting is insufficient, as patients frequently downplayed their symptoms or took them for granted. A proposed questionnaire for primary care, designed to aid in obtaining a detailed medical history, is provided in Appendix A.

Current protocols are lenient regarding the number of symptoms required to diagnose HIT. Our confirmed HIT patients had an average of 38.8/110 positive responses on the questionnaire. It would be worth considering linking the diagnosis of the disease to a minimum score of concurrent symptoms after a larger patient study, alongside a few key symptoms (such as headaches and/or gastrointestinal symptoms). This would greatly help differentiate HIT from other diseases, as not many have such a multifaceted clinical spectrum. Our findings revealed that an average of 4.3 out of the 5 listed organ systems were affected, with 21 symptoms reported out of 62 questions (including miscellaneous complaints). One possible approach could be to provide the questionnaire to patients during their primary care visit to complete at home. In this way, they could monitor their symptoms over an extended period, leading to more accurate responses. It may also be worth considering the inclusion of improvements in higher-stage symptoms as part of the diagnostic criteria for HIT.

The diagnosis could thus be made in six steps after excluding all other diseases. In the first step, suspicion could be established according to the current protocol based on symptoms from two different organ systems. In the second step, the higher the score the patient achieves on the questionnaire, the higher the likelihood of HIT suspicion. In the third step, patients could add their observations to the questionnaire at home and keep a food–symptom diary. Due to the varying frequencies of symptoms, this may require several weeks. In the fourth step, laboratory tests could be considered to further support the likelihood of the disease (e.g., measuring DAO activity). In the fifth step, the diagnosis could be made based on the presence of symptoms in connection with histamine-rich food. This way, we would not exclude patients who prefer not to follow the diet from the diagnosis. The sixth step could involve evaluating improvements after a low-histamine diet with the help of the severity stages, frequency score, and subjective symptom severity scale. The role of primary carers in this process may be important, as they have comprehensive knowledge of the patients’ history.

The willingness of the patients to keep to the diet had a great impact on the effectiveness of this study. The need for a high level of cooperation from patients is a factor that reduces the diagnostic strength of the current HIT diagnostic protocol and may lead to undiagnosed cases. This is also reflected in the predictors of our ROC curve, as patients with more severe symptoms were more willing to participate. Moreover, measuring DAO concentration is not a reliable diagnostic tool; it can only serve as an additional predictor [43,44].

A significant number of patients do not fully adhere to a diet or follow nonvalidated, limiting diets [45]. There is a fine balance to navigate between excluding as few foods as possible while still managing a bearable symptom burden. Ultimately, improvement in patients’ conditions can be achieved through an exclusion diet or a low-histamine diet, with a focus on minimizing limitations in their application [46]. However, suitable therapies are needed in place of dietary restrictions.

### 4.4. Limitations of This Study

This study’s limitations include the relatively low number of patients, the lack of testing for MCAS markers and DAO activity (despite noting that the DAO concentration was low in 82% of the confirmed HIT patients), and the lack of an objective and standardized pre-diagnostic protocol, such as a specialist protocol. Additionally, each patient interview included individualized elements, and a substantial portion of the data were derived from self-reported information. HIT could not be satisfactorily excluded in the subgroup of non-HIT patients because of the absence of an objective diagnostic measure. Further limitations include the potential for selection bias (e.g., the voluntary nature of participation), unmeasured confounding (e.g., environmental factors), residual confounding (e.g., nonstandardized dietary interventions), recall bias (e.g., reliance on memory for specific questionnaire items), Hawthorne bias, misclassification of HIT (e.g., overlooking individual variability in histamine sensitivity or misattributing dietary effects to HIT when other factors such as calorie reduction or changes in eating habits may play a role), and not accounting for multiple comparison testing (e.g., subgroup comparisons within HIT).

## 5. Conclusions

Symptom development in HIT patients appears to follow a specific timeline and presents a stepwise increase in severity stages. Similarly, improvement involves a gradual reduction in both symptom severity and frequency. In the absence of laboratory tests, the established stages and timelines can provide valuable assistance in diagnosing HIT. Indicator symptoms, such as postprandial somnolence, aid in the development of an appropriate diet. A detailed description of symptom conglomerate patterns in general practice is a unique aspect of this research, as is the comparison of analogous features between HIT and MCAD. A six-step diagnostic process has been proposed. After excluding all other potential causes, HIT could emerge as the final diagnosis in a number of cases, serving as a valuable tool for managing symptoms in certain patients.

## Figures and Tables

**Figure 1 jcm-14-00311-f001:**
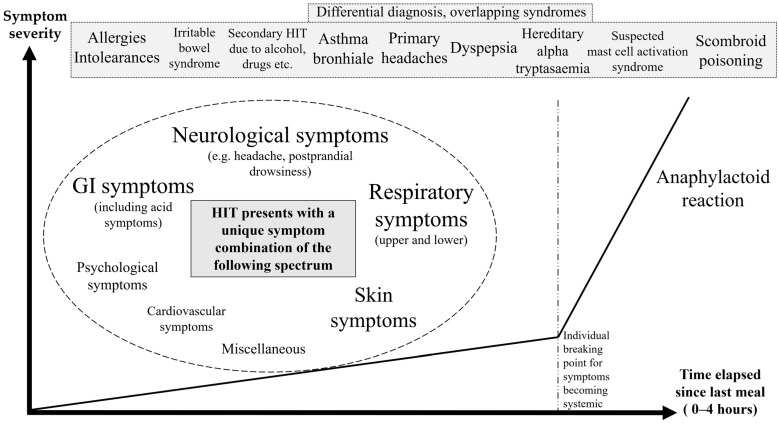
Symptom spectrum of histamine intolerance. The chart was constructed on the basis of various symptom combinations patients encountered, where frequency and severity scores aid in the description and identification of recurring events. Symptoms can be categorized into stages along the severity axis and chronological occurrences along the time axis. In some of these cases, symptoms worsen with the consumption of offending food until they reach an individual breaking point.

**Figure 2 jcm-14-00311-f002:**
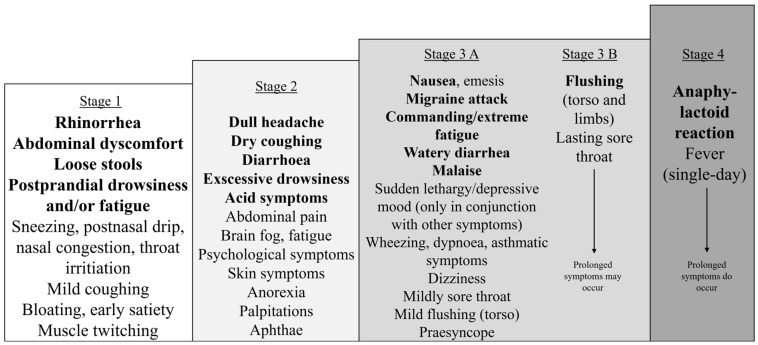
Stages of the most common immediate symptoms of histamine intolerance, organized by severity. The symptoms gradually unfold in a stepwise manner. Each succeeding stage may also include the previous stages. Stages 3B and 4 suggest a tendency toward generalization, with the associated symptoms persisting for a more extended period. An individual may experience various levels of severity after different meals. When a low-histamine diet is applied, patients become symptom-free or symptoms significantly diminish to lower levels.

**Figure 3 jcm-14-00311-f003:**
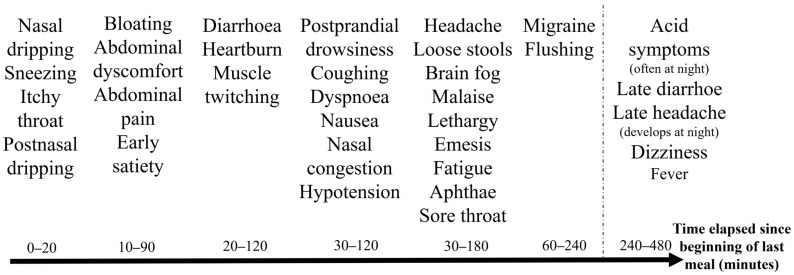
Timeline of immediate symptom development in patients with histamine intolerance following a single meal. Symptoms after a single meal likely correspond to the progression of food through the digestive system, beginning with upper respiratory issues, followed by gastrointestinal symptoms, and concluding with general and neurological complaints. The average time elapsed until the onset of symptoms in patients with histamine intolerance was 1.1 h (SD = 0.9, *n* = 63). The timeline is influenced by both the specific food matrix and the quantities consumed. It complicates recognition that a specific symptom does not always occur at the same time; nonetheless, a characteristic sequence can be observed.

**Figure 4 jcm-14-00311-f004:**
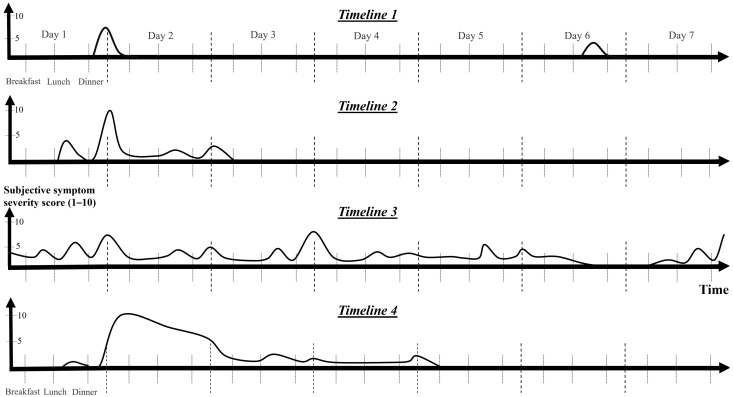
Timelines of potential manifestations associated with histamine intolerance over the course of several days. The timelines were created via personal symptom descriptions gathered from patient interviews as well as from food and symptom diaries. The symptom severity score, frequency score, and duration reports contributed to the development of the charts. Symptoms either appear or worsen after meals. Eighty percent (*n* = 36) of confirmed patients experienced at least three different types of timelines. Isolated episodes with differing intensities (Timeline 1), episodes marked by moderately extended symptoms (Timeline 2), persistent symptoms with intermittent flares (Timeline 3), and severe, prolonged attacks with full-scale symptoms and possible mast cell activation (Timeline 4) are included. As patients begin to implement a low-histamine diet, they gradually move from the more severe end of the spectrum (Timeline 3–4) toward the milder end (Timeline 1) or experience no symptoms.

**Table 1 jcm-14-00311-t001:** Baseline characteristics of all patients and the two patient subgroups.

Baseline Characteristics	All Patients (*n* = 217)	Patients with HIT (*n* = 77)	Patients Without HIT (*n* = 140)	*p* *
Mean age (years)	38.4 ± 12.7 (18–73)	37.8 ± 11.4 (19–68)	38.7 ± 13.4 (18–73)	NS *
Males/Females	29%/71%	31%/69%	29%/71%	
Mean age at symptom onset	22.4 ± 12.7	21.3 ± 11.9	23 ± 13.2	NS *
Onset of symptoms between the ages of 10 and 35 years	175/217 (81%)	63/77 (82%)	112/140 (80%)	
Average duration of symptoms from onset	16.1 ± 13.1	16.6 ± 12.1	15.9 ± 13.7	NS *

HIT = histamine intolerance; NS = not significant; *p* * = comparison of HIT and non-HIT patients with independent T-test.

**Table 2 jcm-14-00311-t002:** The most relevant symptoms of our patients with histamine intolerance (*n* = 77).

Gastrointestinal symptoms		
Bloating	72/77	94%
Diarrhea or loose stools	59/77	77%
Dyspepsia	53/77	69%
Abdominal pain	52/77	68%
Early satiety/discomfort	51/77	66%
Nausea	41/77	53%
Abdominal cramping	37/77	48%
Prone to stomachaches as a child	29/77	38%
Frequent recurrences of aphthae or herpes	28/77	36%
Vomiting	19/77	25%
Neurological symptoms		
Headache	70/77	91%
Fatigue/weakness/weariness	64/77	83%
Postprandial somnolence	62/77	81%
Dull, symmetrical headache	44/77	57%
Migraine or migraine-like (unilateral, throbbing) headache	41/77	53%
Dizziness	41/77	53%
Respiratory symptoms		
Runny nose	52/77	68%
Nasal congestion	46/77	60%
Postnasal drip, throat clearing	44/77	57%
Sneezing	39/77	51%
Sore throat	29/77	38%
Cough	27/77	35%
Asthma or asthmatic symptoms (cough, wheezing, dyspnea, etc.)	16/77	21%
Miscellaneous symptoms		
Skin symptoms	67/77	87%
Psychological	59/77	77%
Muscle twitching	47/77	61%
Cardiovascular (hypotension, presyncope, palpitations)	42/77	55%
Food items that caused complaints in patients with histamine intolerance		
Wine	56/77	73%
Champagne	52/77	68%
Various types of sausages	45/77	58%
Tomato concentrate	42/77	55%
Strong spices	40/77	52%
Dairy products	39/77	51%
Cheese or aged cheese	34/77	44%
Onions	33/77	43%
Citrus fruits	30/77	39%
Coffee	29/77	38%
Chocolate	27/77	35%
Pickles	26/77	34%

**Table 3 jcm-14-00311-t003:** Statistical comparison of all patients, patients with histamine intolerance, and patients without histamine intolerance: (A) according to symptoms, (B) according to the number of complaints, and (C) according to their anamnestic data.

(A)				
Symptoms	All Patients (*n* = 217)	Patients with HIT (*n* = 77)	Patients Without HIT (*n* = 140)	*p* ^a^
Bloating	180 (83%)	72 (94%)	108 (77%)	0.001 ^a^
Diarrhea/loose stools	153 (71%)	59 (77%)	94 (67%)	NS ^a^
Dyspepsia	143 (66%)	53 (69%)	90 (64%)	NS ^a^
Nausea	122 (56%)	41 (53%)	81 (58%)	NS ^a^
Vomiting	59 (27%)	19 (25%)	40 (29%)	NS ^a^
Abdominal pain	119 (55%)	52 (68%)	67 (48%)	0.004 ^a^
Headache	198 (91%)	70 (91%)	128 (91%)	NS ^a^
Migraine-like headache	108 (50%)	41 (53%)	67 (48%)	NS ^a^
Dull headache	133 (61%)	44 (57%)	89 (64%)	NS ^a^
Postprandial somnolence	170 (78%)	62 (81%)	108 (77%)	NS ^a^
Dizziness	106 (49%)	41 (53%)	65 (46%)	NS ^a^
Sore throat	58 (27%)	29 (38%)	29 (21%)	0.006 ^a^
Runny nose	129 (59%)	52 (68%)	77 (55%)	0.048 ^a^
Cough	79 (36%)	27 (35%)	52 (37%)	NS ^a^
Sneezing	110 (51%)	39 (51%)	71 (51%)	NS ^a^
Asthmatic complaints	52 (24%)	16 (21%)	36 (26%)	NS ^a^
Congested nose	121 (56%)	46 (60%)	75 (54%)	NS ^a^
Postnasal drip	104 (48%)	44 (57%)	60 (43%)	0.03 ^a^
Flush	52 (24%)	23 (29%)	29 (21%)	NS ^a^
Brain fog	117 (54%)	43 (56%)	74 (53%)	NS ^a^
Muscle twitching	128 (59%)	47 (61%)	81 (58%)	NS ^a^
Fatigue/weakness	178 (82%)	64 (83%)	114 (81%)	NS ^a^
Aphtha/herpes labialis	66 (30%)	28 (36%)	38 (27%)	NS ^a^
**(B)**				
**Average count of complaints**	**All patients** **(*n* = 217)** **Mean ± SD** **Median (Q1–Q3)**	**Patients with HIT** **(*n* = 77)** **Mean ± SD** **Median (Q1–Q3)**	**Patients without HIT** **(*n* = 140)** **Mean ± SD** **Median (Q1–Q3)**	***p* ^b^**
Average count of gastrointestinal complaints (0–18)	6.3 ± 3 6 (4–8)	6.8 ± 2.8 7 (5–9)	6 ± 3 6 (4–8)	0.024 ^b^
Average count of neurologic complaints (0–9)	4.8 ± 1.6 5 (4–6)	4.9 ± 1.7 5 (4–6)	4.7 ± 1.6 5 (4–5)	0.013 ^b^
Average count of respiratory complaints (0–10)	3.4 ± 2.3 4 (2–5)	4.1 ± 2.1 4 (2–5)	3.6 ± 2.4 4 (2–5)	NS ^b^
Average count of skin complaints (0–8)	1.7 ± 1.4 1 (1–3)	2 ± 1.3 2 (1–3)	1.5 ± 1.4 1 (0–2)	0.009 ^b^
Average count of psychiatric complaints (0–9)	2.5 ± 2.3 2 (1–4)	2.8 ± 2.5 2 (1–5)	2.4 ± 2.2 2 (0.5–4)	NS ^b^
Average count of cardiovascular complaints (0–3)	0.8 ± 0.9 1 (0–1)	0.8 ± 0.1 1 (0–1)	0.7 ± 0.8 1 (0–1)	NS ^b^
Average count of not-tolerated foods (0–26)	6.8 ± 4.8 6 (3–10)	8.9 ± 5 9 (5–12)	5.6 ± 4.2 5 (2–7.5)	<0.001 ^b^
**(C)**				
**Anamnestic data**	**All patients** **(*n* = 217)**	**Patients with HIT** **(*n* = 77)**	**Patients without HIT** **(*n* = 140)**	***p* ^a^**
Feeling more energetic on an empty stomach	138/217 (64%)	51/77 (66%)	87/140 (62%)	NS ^a^
No symptoms during pregnancy (especially in second and third trimesters)	67/71 (94%)	25/26 (96%)	42/45 (93%)	NS ^a^
Onset of symptoms can be attributed to meals	145/217 (67%)	67/77 (87%)	78/140 (56%)	<0.001 ^a^
Onset of symptoms usually in the afternoon or evening	137/217 (63%)	48/78 (62%)	89/140 (63%)	NS ^a^
Children with similar symptoms	43/87 (49%)	18/35 (51%)	25/52 (48%)	NS ^a^
Headaches or chronic diarrhea in the family	144/217 (66%)	54/77 (70%)	90/140 (64%)	NS ^a^
HIT in the family	27/217 (15%)	13/77 (17%)	14/140 (8%)	NS ^a^
Family member who has taken or is taking medication for reflux	99/217 (54%)	39/77 (51%)	60/140 (33%)	NS ^a^

a: Comparison of histamine intolerance patients vs. patients without histamine intolerance, Fisher’s exact test; b: Comparison of histamine intolerance patients vs. patients without histamine intolerance, Mann–Whitney Test; HIT = histamine intolerance; NS = not significant.

**Table 4 jcm-14-00311-t004:** Pre-low-histamine diet (A) and post-low-histamine diet (B) scores.

(A)				
Initial/pre-diet Questionnaire	All Patients (*n* = 217) Median (Q1–Q3) Mean ± SD	Patients with HIT (*n* = 77) Median (Q1–Q3) Mean ± SD	Patients Without HIT (*n* = 140) Median (Q1–Q3) Mean ± SD	*p*
Body mass index (kg/m^2^) (Q1–Q3)	23.2 (20.6–27) 24.5 ± 5	22.6 (20.3–25.7) 23.2 ± 3.9	24.3 (21.1–28.3) 25.2 ± 5.4	0.008 ^a^
Histamine intolerance frequency score (0–5)	3 (3–4) 3.25 ± 1	3 (3–4) 3.5 ± 1	3 (2–4) 3.1 ± 1	0.0025 ^a^
Histamine intolerance symptom severity score (0–10)	8 (6–9) 7.4 ± 1.8	8 (8–9) 7.5 ± 1.9	7 (6–9) 7.3 ± 1.7	NS ^a^
DAO concentration (kU/L)	8 (5.4–10.6) 8.7 ± 5.2	7 (5–9) 7.3 ± 3.2	8.95 (6.85–17.9) 8.1 ± 7	0.015 ^a^
DAO concentration below the specified limit of 10 kU/L	37/50 (74%)	29/35 (81%)	8/15 (53%)	0.03 ^b^
**(B)**				
**Second/post-diet questionnaire**		**Patients with HIT** **(*n* = 77)** **Median (Q1–Q3)** **Mean ± SD**		** *p* **
Histamine intolerance frequency score (0–5) average following dietary adjustments		2 (1–2.75) 1.75 ± 1.3		<0.0001 ^c^
Histamine intolerance symptom severity score average (0–10) following dietary adjustments		3 (1–4.75) 2.9 ± 2.2		<0.0001 ^c^

a: Comparison of patients with HIT vs. patients without HIT, Mann–Whitney test; b: comparison of patients with HIT vs. patients without HIT, Chi-squared test; c: comparison of scores pre- and post-diet, Wilcoxon signed-rank test; HIT = histamine intolerance; NS = not significant.

## Data Availability

The data presented in this study are openly available in Zenodo at https://doi.org/10.5281/zenodo.14042647, reference number 10.5281/zenodo.14042647.

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
