# Peer review of "Spectrum, Time Course, Stages, and a Proposal for the Diagnosis of Histamine Intolerance in General Practice: A Nonrandomized, Quasi-Experimental Study"

_jcm, 2025, doi:10.3390/jcm14020311_

Round 1

Reviewer 1 Report

Comments and Suggestions for Authors

This is a well-written article on the topic of " Spectrum, time course, stages, and a proposal for the diagnosis of histamine intolerance in general practice".  The authors authors are congratulated for conducting such an exhaustive study on histamine intolerance and for having proposed six diagnostic steps for HIT.  However, please note the following comments:

1.  Among the six proposed diagnostic steps for HIT, the fourth step reflects that laboratory tests could be considered to further support the likelihood of the diagnosis.  Given that measuring DAO concentration is not a reliable diagnostic tool and that it can only serve as an additional predictor, please explain what other laboratory tests were included or could be included.  Please comment on the inclusion of a routinely done test - Complete Blood Count (CBC) which could enable calculation of Eosinophil-to-lymphocyte ratio (ELR) which is often considered as a potential indicator.  Other index, although less specific could be the lymphocyte-to-lymphocyte ratio.  Please comment on the validity of these blood cell indices for inclusion in the fourth diagnostic step.  Given that much emphasis has been given on symptomatology representing psychiatric complications, cardiovascular complications, skin, Respiratory, neurological and GIT systems, inclusion of blood cell indices may provide an additional focus in not only diagnosis but also prognosis of patients.  This is just a thought.

2.  Limitations:  Low number of patients, lack of testing for MCAS markers and DAO activity, self-reported information on patient interviews.  Please discuss how the number of patients could be increased, what are possible MCAS markers could be considered or other markers in addition to DAO activity.

Author Response

Subject: Response to Reviewer 1's Comments

Dear Reviewer 1,

We greatly appreciate your constructive feedback on our manuscript. Thank you for taking the time to help improve the quality and clarity of our work.

We have carefully addressed each of the points raised and revised the manuscript accordingly. For your reference, our detailed responses are provided in the attached file.

We hope that these revisions meet your expectations and further enhance the manuscript’s suitability for publication. We remain available for any further revisions or clarifications you may require.

Thank you once again for your valuable feedback.

Best regards,
Dr. József Tamasi

Reviewer 2 Report

Comments and Suggestions for Authors

The manuscript in its current form is comprehensive, interesting

Author Response

Dear Reviewer,

Thank you for taking the time to review our manuscript and for providing such positive review. We are delighted to hear that you found our work suitable for publication. Your comments provide great motivation for our future research.

Should you have any further suggestions or additional feedback in the future, we would be happy to address them.

Best regards,
Dr. József Tamasi

Reviewer 3 Report

Comments and Suggestions for Authors

The authors conducted a quasi-experimental study (a non-randomized study of interventions) to evaluate associations between between food intake and symptoms that might be related to histamine-intolerance (HIT). They present their viewpoints on histamine intolerance and its diagnosis using interesting illustrations and a thoughtful discussion. The topic is important for clinical practice and public health. Thank you for your submission. Suggestions:

1) Please identify in the title and/or in the abstract that this is a nonrandomized, quasi-experimental study

2) Abstract line 12 states "To establish a causal link"- replace this with something like "to evaluate associations of"... This study is not randomized, does not account for potential confounding, and has other methodological limitations which cannot allow one to determine "causality". Therefore, making any statements which indicate causality is incorrect. Please ensure statements indicating causality are not mentioned in the manuscript.

3) Please use the TREND reporting guidelines for arranging this manuscript and ensure all relevant elements in the guideline are reported in the manuscript. Ref: Haynes, A. B., Haukoos, J. S., & Dimick, J. B. (2021). TREND Reporting Guidelines for Nonrandomized/Quasi-Experimental Study Designs. JAMA surgery156(9), 879–880. https://doi.org/10.1001/jamasurg.2021.0552

4) Please clarify if there was any missing data and how it was handled in analyses.

5) A proposed mechanism to explain associations of dietary treatment with potential HIT-related symptoms is influence on gut microbiota. Please elaborate on this in the discussion section; see the following study for example of some evidence (reference/citation):

Sánchez-Pérez, S., Comas-Basté, O., Duelo, A., Veciana-Nogués, M. T., Berlanga, M., Vidal-Carou, M. C., & Latorre-Moratalla, M. L. (2022). The dietary treatment of histamine intolerance reduces the abundance of some histamine-secreting bacteria of the gut microbiota in histamine intolerant women. A pilot study. Frontiers in nutrition9, 1018463. https://doi.org/10.3389/fnut.2022.1018463

6) In section 4.4, please discuss additional limitations of this study such as the potential for selection bias, unmeasured and residual confounding, recall bias, Hawthorne bias; misclassification of HIT, not accounting for multiple comparisons testing, and type I and type II error.

Author Response

Subject: Response to Reviewer 3's Comments

Dear Reviewer 3,

We greatly appreciate your constructive feedback on our manuscript. Thank you for taking the time to help improve the quality and clarity of our work.

We have carefully addressed each of the points raised and revised the manuscript accordingly. For your reference, our detailed responses are provided in the attached file.

We hope that these revisions meet your expectations and further enhance the manuscript’s suitability for publication. We remain available for any further revisions or clarifications you may require.

Thank you once again for your valuable feedback.

Best regards,
Dr. József Tamasi
